# Exploring the Relationship between Sugar and Sugar Substitutes—Analysis of Income Level and Beverage Consumption Market Pattern Based on the Perspective of Healthy China

**DOI:** 10.3390/nu14214474

**Published:** 2022-10-25

**Authors:** Zeqi Liu, Shanshan Li, Jiaqi Peng

**Affiliations:** 1China Institute for Rural Studies, School of Public Policy & Management, Tsinghua University, Beijing 100084, China; 2National Agricultural and Rural Development Research Institute, College of Economics and Management, China Agricultural University, Beijing 100083, China

**Keywords:** sugar-free beverage consumption, sugar-sweetened beverage consumption, income level, health needs

## Abstract

This paper estimates the impact of income level on household beverage consumption, analyzes the consumption trends of sugar-sweetened beverages and sugar-free beverages in households, explores the future changes in the beverage consumption market pattern, and predicts the possible impact of the sugar industry on the development of sugar substitutes based on the beverage consumption data of Kantar Consumer Index in China from 2015 to 2017. The research results show that, firstly, there is an “inverted U-shaped” relationship between income level and household consumption of sugar-sweetened beverages, which indicates that as income rises, household consumption of sugar-sweetened beverages tends to increase and then decrease. Secondly, income level has a positive effect on the household consumption of sugar-free beverages. Finally, in the future stage, with the further growth of income and the promotion of a healthy China, a large amount of sugar substitutes will be added to beverages instead of the original sugar, and the relationship between sugar and sugar substitute consumption will change from complementary to substitution. The findings of this paper have implications for encouraging food and beverage suppliers to produce “healthy”, “nutritious” and “innovative” low-sugar products to meet the health needs of residents and ensure the healthy and orderly development of the sugar industry.

## 1. Introduction

Since the 21st century, obesity has gradually become one of the most problematic public health issues worldwide. The excessive consumption of unhealthy foods high in sugar and calories, especially in the form of sugary drinks, is a major cause of obesity and has been confirmed in several countries with high levels of sugary drink consumption [1,2]. In contrast to these countries, sugary drink consumption in China is growing at a relatively fast rate, despite its low level of consumption. This has also attracted a great deal of attention from the Chinese health authorities. Considering the paradoxical relationship between sugary drink consumption and obesity, the health authorities have accelerated the development of interventions and programs, such as the “Healthy China Action (2019–2030)”. These interventions all emphasize the importance of a healthy diet and recommend replacing sugary beverage consumption with low-sugar or sugar-free beverage consumption to reduce health problems such as obesity [3].

In the process of responding positively to the Healthy China Action, the beverage industry, as the supply side, began to make rapid adjustments to its product formulations and worked hard to innovate various beverage products in line with the concept of nutrition and health. In order to accelerate the implementation of this policy, sugar substitutes such as “aspartame” and “erythritol”, which have almost zero calories but still have a sweet taste, are frequently used in the mass production of sugar-free beverages. Sugar substitutes are quickly becoming a new trend in the beverage industry, replacing existing sugar in beverages, and creating new opportunities. Behind this is a willingness to consume on the demand side.

What are the factors that influence the consumption of beverages on the demand side? According to economics, income is one of the main factors affecting the demand for various types of products. Studies have shown that the increase in income level will drive residents to increase the demand for healthier food with higher nutritional content [4,5]. It is thus worth considering whether the rising income level of residents has led to the relaxation of budget constraints, and whether this prompts residents to increase their demand for healthy products such as sugar-free beverages and accelerate the reduction of residents’ demand for sugar-sweetened beverages. Will the change in demand for healthier products change the original pattern of the beverage consumption market and prompt the beverage industry to significantly adjust the sugar (cane sugar) formula? As the beverage industry is the industry that uses the most sugar, its development trend is closely related to the sugar consumption and production chain. Therefore, will the change of the beverage consumption market pattern change the relationship between sugar consumption and sugar substitutes, thus affecting the development of China’s sugar industry?

Studies have shown that in the early 21st century, rising incomes have driven a shift in the structure of sugar consumption in China [6,7]. During this period, as income levels rose, residents were no longer satisfied with the single calorie requirement provided by direct sugar, such as white sugar and brown sugar, and the consumption of sugary drinks, such as carbonated beverages, which provide a “sweetener function”, increased rapidly, and quickly took over most of the beverage consumption market. At the same time, this has contributed significantly to the growth of sugar consumption in China [8,9,10,11].

However, at this stage, it is worth exploring whether the rapid development of sugar substitutes, as one of the common ingredients in health products, in beverages or other food areas will disrupt the existing pattern of sugar consumption in China. Some scholars have explored the relationship between starch sugar consumption and table sugar consumption, but the conclusion is still controversial. Studies have shown that 50% of sugary beverage varieties contain starch sugar in their formulations, and the scale is expanding, and starch sugar in carbonated beverage formulations has largely achieved complete substitution for white sugar [12]. It is evident that starch sugar, as one of the sugar substitutes, will continue to crowd out part of the market for sugar consumption in China [13], and may change the situation whereby table sugar has dominated the sugar consumption market for many years, making the sugar market more vulnerable [14]. However, some scholars believe that despite the rapid development of starch sugar in recent years, it is still a relatively small industry compared with table sugar, and the market competitiveness of starch sugar will continue to weaken due to the cost, giving way to partially crowding out the market share of table sugar consumption [15].

Some of the literature provides references and directions for the content of this paper, but it still cannot answer the questions of this paper. Firstly, few studies have analyzed the impact of residents’ income level on the beverage consumption market pattern and conducted quantitative analyses. Secondly, few scholars have also considered the current residents’ demand for health products and split the beverage consumption market into two, analyzing the consumption trends of sugar-free beverages and sugar-containing beverages separately from the demand side. Finally, the use of sugar substitutes in the beverage market in recent years is gradually crowding out the sugar consumption market, and the beverage consumption market, as the largest sugar (sucrose) consumer, can influence the changes of sugar consumption in China to some extent. Based on this, the main task of this paper is to analyze the impact of income level on the demand for sugar and sugar-free beverages from the perspective of a healthy China, and to explore the possible changes of the beverage consumption market pattern in the future. In addition, by studying the development trend of sugar-free beverage consumption in the beverage industry, this paper will also predict the possible impact of the sugar industry due to the development of sugar substitutes. From this perspective, this paper also aims to provide direction and inspiration for future policy adjustments in the beverage consumption market and the development of the sugar industry.

## 2. Materials and Methods

### 2.1. Data Source

The data used in this paper come from the 2015–2017 China Household Beverage Consumption Dynamics Tracking Survey (Kantar Worldpanel) by the Kantar Consumer Index, which provides insight into changes in Chinese household beverage consumption behavior and provides a reliable source of data for academic research and public policy analysis, while better supporting the entire analytical framework and content of this paper. The data sample covers 24 provinces, with a target sample size of 181,613 households, and the survey only includes respondents in the sample households. The data used in this paper, with 2015 as the baseline survey data, contains 31,175 households with a total of 31,175 respondents(annual rotation of sample of approximately 25%), of which, 3315 are male and 27,860 are female. The data contain daily and annual samples of total beverage household consumption, number of individual beverages consumed, and package size, and the beverage categories are divided into sugary and sugar-free beverages. Among them, sugary beverages include six types of sugary ready-to-drink coffee, sugary carbonated drinks, sugary fruit juices, sugary functional drinks, sugary Asian traditional drinks, and sugary ready-to-drink tea. Correspondingly, sugar-free beverages include sugar-free ready-to-drink coffee, sugar-free carbonated beverages, sugar-free fruit juices, sugar-free functional beverages, sugar-free Asian traditional beverages and sugar-free ready-to-drink teas in a total of six categories. In addition, the data also contained the following two sections, basic household information (household income, household form, household type, presence of children in the household) and information on the demographic characteristics of the respondents (age, gender, education level) [16].

### 2.2. Variable Definition

The explanatory variable in this paper is household beverage consumption, which is further subdivided into household sugar-sweetened beverage consumption and household sugar-free beverage consumption, and is mainly used to measure the beverage consumption of residential households. Although there may be substitution effects between the various types of beverages, this paper ultimately aims to explore the possible changes in future consumption demand for total sugar-sweetened beverages and sugar-free beverages and the relationship between the two. Therefore, this paper defines the explanatory variable, household consumption of sugary beverages, as the sum of the consumption of sugary ready-to-drink coffee, sugary carbonated beverages, sugary fruit juice, sugary functional beverages, sugary Asian traditional beverages, and sugary ready-to-drink tea, in order to investigate the total consumption of sugary beverages in households. Household consumption of sugar-free beverages was defined as above.

The core explanatory variable in this paper is total household income. Referring to the existing studies [17,18], the control variables were set as household size, age and gender of the respondents, and education level. In addition, the presence or absence of children in the household was also included as a control variable in this paper, because children tend to eat the food that their parents eat and their eating habits and behaviors are influenced by family members, therefore, families with children are generally more health conscious and will try to reduce the consumption of unhealthy foods such as sugary drinks to avoid children being affected by them [19,20].

### 2.3. Emperical Test

Since the data used in this paper are unbalanced panel data, and it is necessary to observe variables that do not change over time such as gender and province, the model estimation method needs to be chosen between random effects estimation and mixed effects estimation. Using the LM test, the original hypothesis of “individual random effects” is accepted, so the random effects model is chosen, and the parameters are estimated using the feasible generalized least squares estimation method (FGLS). The model is set as follows.
(1)$$Y_{SSB}=\theta_{0}+\theta_{1}Inc+\theta_{2}lnInc^{2}+\theta_{3}Z+\theta_{4}R+\varepsilon_{1}$$
(2)$$Y_{NSSB}=\alpha_{0}+\alpha_{1}Inc+\alpha_{2}lnInc^{2}+\alpha_{3}Z+\alpha R+\varepsilon_{2}$$
where $Y_{SSB}$ and $Y_{NSSB}$ are the household consumption of sugar-sweetened beverages and household consumption of sugar-free beverages, respectively. lnInc is the logarithm of household income; Z is the household characteristics variable and respondent characteristics variable, including household size, presence of children, respondent age, respondent gender, and respondent education level. $\theta_{0}$ … $\theta_{4}$, and $\alpha_{0}$ … $\alpha_{4}$ are unknown parameters. $\varepsilon_{1}$ and $\varepsilon_{2}$ are random errors. The price variables are not considered in the model, mainly because the beverage consumption involved in the article contains various types of different sugar-sweetened and sugar-free beverages, which are extremely different and difficult to account for, so it is assumed that residents of the same province face the same beverage market price during the survey period, and the province dummy variable can control the impact of price differences on consumption between provinces to some extent [21]. Therefore, a province dummy variable R was introduced in the model.

### 2.4. Statistical Analysis

Since the data used in this paper are unbalanced panel data, and it is necessary to observe variables that do not vary over time such as gender and province, the model estimation method needs to be chosen between random effects estimation and mixed effects estimation. Using the LM test, the original hypothesis of “individual random effects” is accepted, so the random effects model is chosen, and the parameters are estimated using the feasible generalized least squares estimation method (FGLS). Models 1 and 3 include only the household income variable and the squared household income variable, while models 2 and 4 add control variables and province dummy variables. Considering that the high correlation between Inc and $Inc^{2}$ may lead to the existence of multiple covariances in the model, therefore, this paper refers to the previous method [22], which will use a centralized treatment, lnInc minus its own mean treatment before constructing the squared term, to eliminate or reduce the problem of multiple covariances (variance inflation factor VIF < 5). All analyses were performed using stata15 statistical software, and *p*-values < 0.1 were considered statistically significant.

## 3. Results

### 3.1. Basic Characteristics of Sample

As shown in Table 1, the average annual household consumption of sugary drinks in the sample was 28.55 L, and the average household consumption of sugar-free drinks was 0.98 L. The average household income was 100,500 yuan per year, and they were generally families of three, with approximately 28% of the households in the sample having children. The average age of the respondents was 44 years old, and 44.23% were female and the average education level was above high school.

### 3.2. Basic Regression Analysis of Household Income on Beverages

Models 1 and 3 only contain the logarithm of household income and its squared term, while models 2 and 4 add a series of control variables on top of models 1 and 2, and the resulting estimation results are shown in Table 2 and Table 3.

First, from the regression results, the estimated coefficients of the log of household income in both model 1 and model 2 are significantly positive, and the coefficients of the squared terms of the log of income are significantly negative, indicating that there is a nonlinear effect of household income on household consumption of sugary drinks, which is more common in food category studies [23,24]. On the one hand, the marginal effect of household income on household consumption of sugary beverages is not constant, and the latter varies with the former. The comparison of coefficients in model 2 shows that the absolute value of the logarithmic coefficient of income, i.e., the coefficient of the linear relationship, is 0.086, which is larger than the squared term of 0.034, indicating that the linear relationship dominates when the income is small, and the coefficient of the squared term will gradually dominate as the income keeps increasing. On the other hand, the relationship between household income and household consumption of sugary drinks may have an inverted U-shaped relationship, but this relationship depends on whether the turning point falls within the sample interval. This means that household consumption of sugary drinks does not always increase with income, but may fall after income reaches a certain level. Regardless, further verification is needed.

In this paper, the marginal effect of household income on household consumption of sugary drinks is calculated to verify whether there is an “inverted U-shaped” relationship between the two, and the results are shown in Figure 1. When the logarithm of income reaches 12.9, which means when the household income is 400,300 yuan, the marginal effect is 0. At the same time, the marginal effect shows a positive to negative characteristic here. Thus, the existence of the “inverted U-shaped” relationship is confirmed, implying that household consumption of sugar-sweetened beverages increases with household income in the early stage, but then tends to fall with income growth after reaching the turning point (at the household income of 400,300 yuan). 

Second, the effect of log household income on household sugar-free beverage consumption is significantly positive in Models 3 and 4, but only the coefficient of the squared term of log income is significantly positive in Model 4. Overall, an increase in household income has a positive incremental effect on household sugar-free beverage consumption, which represents as household income increases, household sugar-free beverage consumption shows an exponential increase. Meanwhile, combined with Figure 2, it is found that the samples in this paper all fall within the interval where the marginal effect is positive, again proving that the increase in household income has a boosting effect on household sugar-free beverage consumption.

In terms of other variables, there is a significant positive correlation between household size and household consumption of sugary beverages and a significant negative correlation between household consumption of sugar-free beverages, i.e., a one unit increase in household size is associated with a 0.113 unit increase in household consumption of sugary beverages and a 0.20 unit decrease in household consumption of sugar-free beverages. There is a significant negative correlation between the presence of children in the household and the household consumption of sugary drinks, i.e., households with children consume 0.068 units less sugary drinks than households without children. The age of the respondent has a significant negative correlation with household consumption of sugary drinks and a significant positive correlation with household consumption of sugar-free drinks, i.e., an increase of one unit in the respondent’s age reduces household consumption of sugary drinks by 0.294 units and increases household consumption of sugar-free drinks by 0.273 units. There was a positive correlation between the gender of the respondent and household consumption of sugary beverages and household consumption of sugar-free beverages, with men consuming more beverages. There was a significant negative correlation between the education level of the respondent and household consumption of sugary beverages, i.e., one unit increase in education level of the respondent was associated with a 0.028 unit decrease in the household consumption of sugary beverages.

### 3.3. Robustness Test

The above regressions suggest that household income has an “inverted U-shaped” relationship with household consumption of sugar-sweetened beverages and an exponential relationship with household consumption of sugar-free beverages, but the robustness of the findings needs to be further verified.

In fact, households make rational choices of beverages based on their own endowment characteristics, which means choosing sugary or sugar-free beverages or not is a kind of “self-selection” behavior of households. Ignoring this self-selection problem and directly estimating the effect of household income on household beverage consumption may lead to biased results. Therefore, this paper uses the Generalization Propensity Score (GPS) matching method, which is suitable for handling multi-valued variables, to address the self-selection bias of household beverage consumption and to determine the robustness of the findings. The propensity score matching method is not chosen in this paper because it is only suitable for solving the problem of binary variables with 0 or 1 and can only obtain the average treatment effect of household income on household beverage consumption, which cannot portray the dynamic changes of household beverage consumption at different income levels and may obscure or underestimate the actual effect between the two.

The GPS approach requires that the assumption of conditional independence between the treatment and outcome variables be satisfied holds that:(3)$$Y\left(t\right)⊥T|X\;(\forall t\in \left[t_{0}\;,\;\mathrm{the}t_{1}\;\right)$$
where T is household income, the treatment variable, and Y(t) is the magnitude of household beverage consumption corresponding to when the treatment variable household income takes the value of *t*, the outcome variable. For the treatment variable to satisfy T ∈ [0, 1], this paper draws on a previous research method [25,26], which further defines the treatment variable as the ratio of the ith household income to the income of the highest-income household by excluding individual outliers based on the distribution of household income levels. The variables contained in the vector X are called “matching variables”, also known as covariates, indicating control variables that can affect both the outcome variable Y and the treatment variable T. The covariates in this paper will be selected to control for household size, the presence of children in the household, the age of the respondent, the gender of the respondent, the education level of the respondent, and province.

The GPS method is implemented in three steps. In the first step, the conditional probability density distribution of the treatment variable T(t) is estimated given the covariate X. Since household income in the sample presents a non-normal distribution, the conditional probability density of household income is estimated by choosing the Fractional Logit model in this paper. In the second step, the outcome variable Y(t) is expressed as a function of the treatment variable T and the generalized propensity score variable, and its expectation condition is estimated using the OLS method. In the third step, the average expectation of the outcome variable Y at *t* for the treatment variable T is estimated. The results of the first and second steps are shown in Table 4, and the overall results are as expected.

In the third step, the paper divides the values of the treatment variables into five regions and estimates the causal effect of household income on household beverage consumption for each of the five regions. The causal effects of different household incomes in the five regions are linked to obtain the “dose-response” between household income and household beverage consumption in the whole range of treatment variables. The two images in Figure 3 verify the relationship between household income and the consumption of sugary and sugar-free beverages, respectively, and a significant difference can be observed between the two. As shown in the left panel of Figure 3, the relationship between household income and household consumption of sugary beverages is not linear, and as household income increases, household consumption of sugary beverages shows an “inverted U-shaped” curve that rises first and then falls. As shown in the right panel of Figure 3, the relationship between household income and household consumption of sugar-free beverages is not linear, and the household consumption of sugar-free beverages grows logarithmically with the increase of household income. Therefore, the conclusions of this paper are robust.

## 4. Discussion

In the current study, we found that household income levels have a significant impact on household beverage consumption, and that further increases in income levels will lead to more households choosing sugar-free beverages and gradually reducing their consumption of sugary beverages in the future. The reason for this is that for households with higher incomes, they have healthier diets and healthier lifestyles and eating habits, and therefore will increase their consumption of nutrient-rich foods and decrease their intake of carbohydrate foods [27,28,29]. While sugary beverages, as one of the representatives of carbohydrate foods, have become an inevitable trend to reduce the consumption with the big step of household income [30,31]. This means that to a certain extent, this will undoubtedly change the landscape of the beverage consumption market, while the continued development of the sugar substitute industry is also highly likely to influence the future development of the sugar industry.

Overall, household consumption of both sugary and sugar-free beverages will grow with income when household income is less than 400,300 yuan. The average annual income of Chinese resident households in 2020 will be about 96,600 yuan, an increase of 4.7% from 2019. Taking 2019 as the base year and assuming a constant annual income growth rate, the average annual income of Chinese resident households in 31 years will exceed 400,300 yuan. This means that sugar consumption in terms of sugar for beverages will not decrease significantly over the next 30 years due to the advent of sugar substitutes, but its growth rate will be lower. In a sense, the development of sugar substitutes is complementary to the supply of table sugar during this period, and the relationship between the two is more like a complementary relationship than a complete substitution. This is in line with the findings of Si Wei and Zhu Haiyan [14]. However, after household income exceeds 400,300 yuan, household consumption of sugar-sweetened beverages begins to decline, while at the same time, sugar-free beverage consumption continues to have good room for growth. This means that in the coming period, the emerging sugar-free beverage industry will weaken the market competitiveness of the traditional sugar-containing beverage industry, and the dominant position of sugar-containing beverages in the beverage market will be broken by sugar-free beverages, and the beverage consumption market pattern will be greatly changed or even reshaped. In addition, the beverage industry, as the industry that uses the most sugar, the continued decline in the consumption of sugar-sweetened beverages has led to a significant reduction in sugar consumption, which also means that the relationship between sugar and sugar substitutes will also shift from complementary to substitution.

We also found that at least three of the control variables carried behind them the population’s need for a healthy diet. First, the presence of children coefficient is significantly negative, implying that the consumption of sugary beverages decreases in households with children, which is in line with expectations. Some argue that diet has a memory function, and that eating habits during childhood remain for life [32,33]. Therefore, healthy eating habits during childhood will be maintained for a long time. Then, less exposure to and intake of unhealthy foods such as sugary drinks during childhood would mean that consumption of sugary drinks would decrease in adulthood. Second, there is a significant negative relationship between respondents’ age and the household consumption of sugary beverages, while there is a significant positive relationship with the household consumption of sugar-free beverages. This indicates to some extent that there is a stratification of beverage consumption among residents of different age groups, especially for middle-aged and elderly people, less sugar, less salt and less oil are important principles of their food consumption [34]. Therefore, it is a more reasonable choice for this group to reduce sugary beverage consumption and increase sugar-free beverage consumption. Third, the effect of respondents’ education level on household beverage consumption is similar to respondents’ age, and there is a stratification of beverage consumption among residents with different education levels. In general, residents with higher education levels are also more health conscious and their health needs are greater [35,36]. Thus, this group will reduce sugary beverage consumption.

At the same time, based on the results of the study, we speculate that the intrinsic health needs of the population are an important factor in the increasing consumption of sugar-free beverages, which in turn leads to changes in the beverage consumption market pattern, but they need to be stimulated by certain conditions, such as an increase in income levels, to function better. Studies have shown that only those with higher income levels and sufficient economic conditions as supports can purchase healthier and nutritious food and make lifestyle choices that are beneficial to their health, thus satisfying their health needs and pursuing healthy behaviors [37,38]. As the most representative health product, sugar-free beverages can not only ensure the need for beverage taste, but also carry the health needs of residents, and with the increase of income level, it is only in contempt to become the best substitute for sugary beverages. This also indicates that in the general environment of building a healthy China, with the growth of income, the residents’ consumption of sugar or beverages changes from a single demand for sweeteners to a health-oriented demand with sweeteners as a supplement is likely to become a popular trend in the future.

Finally, considering that a sugary beverage tax, one of the most cost-effective public health strategies, is a common price instrument used in several countries to tackle obesity [39,40], it may also be one of the intervention policies to tackle rising sugary beverage consumption in China in the future. It has been argued that a tax on sugary beverages would induce a shift in the consumption of sugary beverages to sugar-free or low-sugar beverages among the population [41,42]. Then, with the gradual increase in income levels and nutritional health awareness of the population, coupled with the sugary beverage tax, we can imagine that the consumption of sugary beverages by the population will further decrease significantly, thus accelerating the change in the beverage consumption market pattern.

Our study has the following strengths: firstly, there is a lack of current research related to the emerging beverage consumption market in China, which this paper complements. Secondly, the paper uses the GPSM approach to deal with possible self-selection issues, while also addressing possible endogeneity issues and confirming the robustness of our results. However, there are some limitations to this study. Due to data limitations, there is a lack of variables on nutritional perceptions and food preferences, and we are unable to conduct mechanistic tests or better disentangle the direct and mediating effects of income. Overall, this study has served as a primer and will hopefully provide some inspiration for subsequent research in this area. We will also follow up on this study and organize research on the relevant variables to broaden the depth and breadth of the study on this basis and continuously improve it to make up for the shortcomings.

## 5. Results

In conclusion, our findings show that there is an inverted U-shaped relationship between household income and household consumption of sugary beverages, and an exponential relationship with the household consumption of sugar-free beverages. This means that once household income reaches a certain point, further increases will drive household sugary beverage consumption down, while sugar-free beverage consumption will continue to rise. The findings of this paper have implications for the beverage consumption market and the sugar industry. With the growth of income and the continued promotion of a healthy China, consumers’ own nutritional awareness has been strengthened, making it highly likely that the consumption demand for sugary foods, including sugary beverages, will decrease. This is not only related to the reshaping of the beverage consumption market pattern, but also has a real impact on the future development of sugar consumption in China. If the supply side of sugar is not adjusted quickly, it may cause some industrial shocks. Firstly, Chinese sugar enterprises and related departments should encourage the food and beverage industry to transition from concentrating resources on the production of high-sugar goods to the production of “healthy”, “nutritious” and “innovative” low-sugar goods to meet the current and future needs of the population. Secondly, the food and beverage industry should be encouraged to make use of technological innovation to lower the cost of low-sugar goods and improve the competitiveness of prices.

## Figures and Tables

**Figure 1 nutrients-14-04474-f001:**
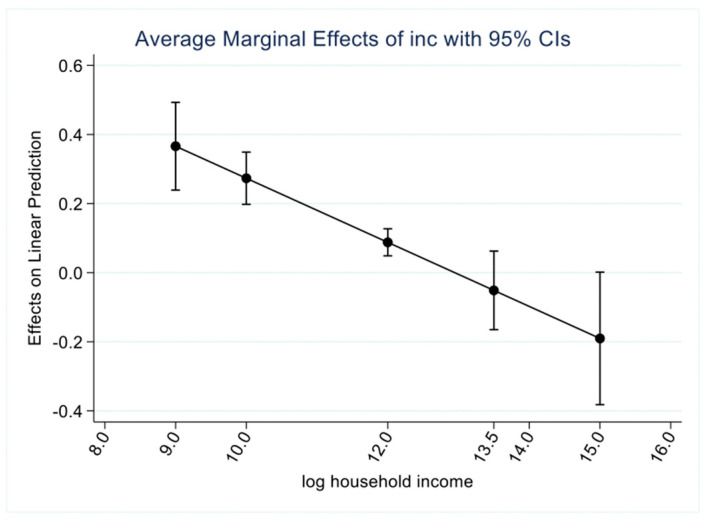
Average marginal effect of income on the household consumption of sugary drinks.

**Figure 2 nutrients-14-04474-f002:**
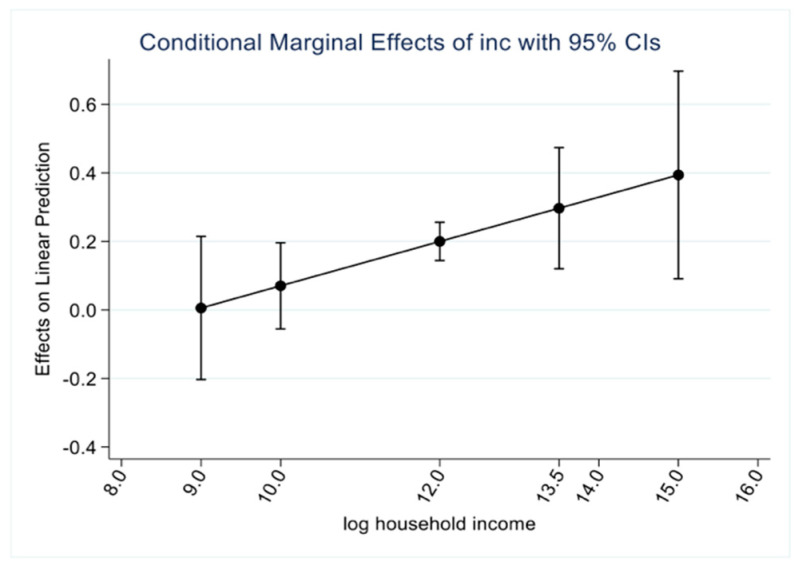
Average marginal effect of income on the household consumption of sugar-free drinks.

**Figure 3 nutrients-14-04474-f003:**
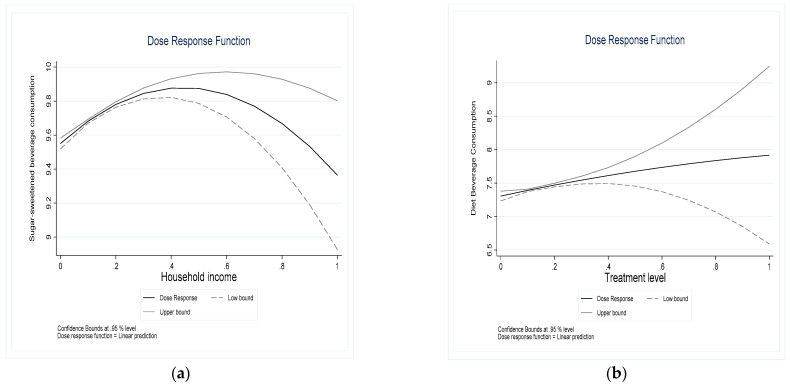
“Dose response” function of different household incomes on household beverage consumption. (**a**): A “dose-response” function of household income and household consumption of sugary drinks; (**b**): A “dose-response” function of household income and household consumption of sugar-free drinks.

**Table 1 nutrients-14-04474-t001:** Basic characteristics of sample.

Variable	Definition	Mean	S.D.	Minimum	Maximum
Household consumption of sugary drinks	liters/year	28.55	35.83	0	1082.97
Household sugar-free beverage consumption	liters/year	0.98	3.78	0	200.49
Household income	10,000 yuan/year	10.05	4.93	1.48	72.00
Family Size	number of family members	2.77	1.11	1	9.00
Availability of children	dummy; 0 = no; 1 = yes	0.28	0.49	0	1
Respondent Age	age measures by year	44.23	11.51	14.00	91.00
Gender of the respondent	dummy; 0 = female; 1 = male	0.11	0.31	0	1
Respondents’ education level	educational years	11.79	2.52	0	19.00

**Table 2 nutrients-14-04474-t002:** Effect of household income on consumption of sugary drinks.

Variables	Model 1 ^a^	*p*-Value	Model 2 ^b^	*p*-Value
Log of household income	0.148 ***^c^	0.000	0.086 ***	0.000
	(0.011)	<0.01	(0.014)	<0.01
Square of the logarithm of household income	−0.034 **	0.011	−0.034 **	0.027
	(0.013)	<0.05	(0.015)	<0.05
Family size			0.113 ***	0.000
			(0.007)	<0.01
Availability of children			−0.068 ***	0.000
			(0.014)	<0.01
Respondent’s age			−0.294 ***	0.000
			(0.027)	<0.01
Gender of the respondent			0.146 ***	0.000
			(0.021)	<0.01
Respondents’ education level			−0.028 ***	0.000
			(0.003)	<0.01
Province dummy variables			Control	
Constant	9.690 ***		10.618 ***	
	(0.006)		(0.121)	
$R^{2}$	0.004		0.038	
Sample size	91,826		71,190	

^a^ Model 1 is the test results obtained by regressing household income and sugary drinks consumption using the FGLS approach without the inclusion of control variables. ^b^ Model 2 is the test results obtained by regressing household income and sugary drinks consumption using the FGLS method after adding control variables and controlling for the dummy variables of the provinces. ^c^ This regression was analyzed using Stata15 statistical software, and a *p* < 0.1 is statistically significant. Where “**” represents 0.01 < *p* < 0.05, and “***” represents *p* < 0.01. The smaller the *p*-value, the better the performance.

**Table 3 nutrients-14-04474-t003:** Effect of household income on consumption of sugary-free drinks.

Variables	Model 3 ^a^	*p*-Value	Model 4 ^b^	*p*-Value
Log of household income	0.147 ***^c^	0.000	0.134 ***	0.000
	(0.016)	<0.01	(0.024)	<0.01
Square of the logarithm of household income	0.027	0.234	0.043 *	0.076
	(0.022)	-	(0.025)	<0.1
Family size			−0.020 *	0.054
			(0.011)	<0.1
Availability of children			−0.022	0.337
			(0.023)	-
Respondent’s age			0.273 ***	0.000
			(0.039)	<0.01
Gender of the respondent			0.072 **	0.010
			(0.030)	<0.01
Respondents’ education level			0.006	0.146
			(0.004)	-
Province dummy variables			Control	
Constant	7.384 ***		6.332 ***	
	(0.009)		(0.178)	
$R^{2}$	0.005		0.056	
Sample size	26,252		19,959	

^a^ Model 3 is the test results obtained by regressing household income and sugar-free drinks consumption using the FGLS approach without the inclusion of control variables. ^b^ Model 4 is the test results obtained by regressing household income and sugar-free drinks consumption using the FGLS method after adding control variables and controlling for the dummy variables of the provinces. ^c^ This regression was analyzed using Stata15 statistical software, and a *p* < 0.1 is statistically significant. Where “*” represents 0.05 < *p* < 0.1, “**” represents 0.01 < *p* < 0.05, and “***” represents *p* < 0.01. The smaller the *p*-value, the better the performance.

**Table 4 nutrients-14-04474-t004:** Generalized propensity score matching first and second step regression results ^a^.

Step 1.	Fractional Logit		Step 2.	OLS
Variables	Y = household income	Variables	Y = Household beverage consumption	
			Household consumption of sugary drinks	Household sugar-free beverage consumption
Family Size	0.235 *** (0.002) ^b^	T	1.804 *** (0.277)	0.019 (0.497)
Availability of children	−0.112 *** (0.005)	T^2^	−1.680 *** (0.359)	−0.258 (0.912)
Respondent’s age	0.175 *** (0.007)	R	−1.002 * (0.581)	6.010 *** (1.074)
Gender of the respondent	−0.048 *** (0.006)	R^2^	7.827 *** (2.226)	−19.057 *** (4.304)
Respondents’ education level	0.048 *** (0.001)	T * R	−2.250 (1.842)	6.147 (3.791)
Province dummy variables	Control	Cons	9.524 *** (0.041)	6.874 *** (0.074)
Constant	−3.810 *** (0.035)	F-value	92.520	32.000
AIC	0.566	Decision factor	0.070	0.077

^a^ Table 4 shows the test analysis using GPSM, where T is the treatment variable, R is the GPS score, and T * R is the interaction term between the two. ^b^ “*” represents 0.05 < *p* < 0.1, and “***” represents *p* < 0.01. The smaller the *p*-value, the better the performance.

## Data Availability

Not applicable.

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
