# Peer review of "Exploring the Relationship between Sugar and Sugar Substitutes—Analysis of Income Level and Beverage Consumption Market Pattern Based on the Perspective of Healthy China"

_nutrients, 2022, doi:10.3390/nu14214474_

Round 1
Reviewer 1 Report
This is a straightforward investigation of beverage consumption and income in China, showing an inverted u-shaped relation between income and consumption of sugar-sweetened beverages. I have no major concerns about the paper, in general, although I do think it could benefit from copyediting -- in places there are some oddly worded sentences, but these are not frequent. Good luck with your research!
Author Response
Dear reviewer,
Thank you very much for your acknowledgement and support of our paper and for giving us this opportunity to have our article discovered by a wider audience. The suggestions you made were also very pertinent, and based on what you said about some oddly worded sentences in the text, we have also revised the whole text throughout to better help the reader understand it. Thank you again for your guidance and support for this paper!
Reviewer 2 Report
Based on the beverage consumption data obtained from the Kantar Consumer Index in China from 2015 to 2017, the authors of the present manuscript aim to estimate the impact of the parameters affecting the beverage consumption in households, focusing the attention on sugar-sweetened vs sugar-free beverages.
The results of the research suggest an inverted trend between the income level and the consumption, so that the income rises, and household consumption of sugar-sweetened beverages tends to increase and then decrease. This finding joined to the healthy China program ensures, in their opinion, an innovative and healthy development of sugar industry. Showing a quite positive spirit, the authors are confident that in future stage we will see a further growth of the income of Chinese population so that the tendence in the beverage consumption too will follow a promising trend towards the sugar free choice.
Despite the interest of the matter, in this reviewer’s opinion, the manuscript is, in this version, just unreadable. Dealing with a scientific paper, the reader must often make a great effort in trying to understand whether the sentence is reporting results or the authors are just talking about the matter of the research without any experimental evidence. Indeed, throughout the work, the authors present mathematical models and tests supporting their findings, but the prolix and tedious exposition leads to a great confusion.
Finally, most important remark, the author’s results need further considerations.
Starting from these general considerations, this reviewer strongly suggests a deep revision as follows:
1. The manuscript must be shortened and all the sentence and remarks that do not strictly arise from the data of the study must be eliminated. Also the Introduction is too long, containing a lot of general sentence suitable for a magazine and not for a scientific journal.
2. Figures and Tables must be linked to the text and at the same time the Captions much more explicative.
3. Overall, the criticism raised by this reviewer concerns the results of the study. In fact, the authors ascribe to the household income, even grouped in different income brackets, the propensity to consume sugar free beverages, and this is the only drastic conclusion of the study. How is this variable significantly related to the cultural level of the family? How to the age? As previously reported, the authors are positively confident that in the next years the Chinese society is destined to a further growth of the income so that the tendence in the beverage consumption too will follow a promising trend towards the sugar free choice. Thus, the authors individuate a link between the income and the Chinese health policy. Which the role of an improvement in the cultural level? The authors must clarify these points. At the same time, Tables and Figures must be easily understandable, self-explaining and deeply connected to the text.
In particular
1. Line 143 which the reason of the great difference between the number of respondent males and females?
2. Line 209 The Table doesn’t report the statistical difference of the data.
3. Figure 1 The choice of the symbols in the Figure makes it unclear. Again, the statistics is not reported.
4. Eq. 1 and 2. The variables must be indicated following the mathematical style and not throughout the text.
5. The Result section must be shortened and focused only on Tables and Figures. At the same time the Captions must be enriched of explaining comments to help and guide the reader.
6. The Discussion Section is meaningless. In a scientific paper the Discussion is the section where the authors discuss their results considering previous findings reported by the literature. Only one reference is reported in the Section.
7. The limitations of the study must be reported possibly in a separate section.

Author Response
Dear reviewer,
We appreciate your constructive comments and suggestions on our manuscript. We have revised the manuscript based on your comments and the revised manuscript has been uploaded to the website. Because the revision note is too long, we have placed it in the attachment. Please see the attachment.

Reviewer 3 Report
Dear Authors,
Even if the subject is interesting, there are several flaws concerning the expression of data in Tables and discussion of results.
The Introduction should be reorganized and rewritten to better focus the aim of manuscript.
Data reported in Tables 1, 2, 3 and 4 are not clear. Check them and explain the source of data. Consequently the description and discussion of related results is difficult to follow for the reader and it must be clarified and implemented.
Also the statistical analysis reported in Table 3 is not clear as well the sample size.
Figure 1 is not clear and not correspond to the text (lines 201-208).
The discussion of results in Figure 2 and Figure 3 should be improved.
The format of paper should be set up following Journal guidelines
Author Response

(The authors gave the same response as above.)

Round 2
Reviewer 2 Report
After the authors' revision, I think the manuscript in this revised version suitable for publication in Nutrients journal.
Author Response
Dear reviewer,
Thank you very much for your constructive comments on our manuscript in the last round. We learned a lot from the last round of revisions and agreed that the manuscript has been greatly improved. We would like to express our sincere gratitude to you.
Reviewer 3 Report
Dear authors,
I found the manuscript improved over the previous version.
Please check carefully the manuscript, delete the note in the text, add the reference used for the the paragraph 2.1, Data source.
Add a paragraph for the statistical analysis used, and explain the statistical output in the tables note.
Please, check the tables carefully, the table notes, and make the tables self-explanatory (now they are not).
Best regards
Author Response
Dear reviewer,
We really appreciate your suggestions to our manuscript in the second round of review, and we have carefully revised the manuscript based on your comments. Because the revision note is too long, we have placed it in the attachment, please see the attachment.
king regards
